# Novel Insights in the Physiopathology and Management of Obesity-Related Kidney Disease

**DOI:** 10.3390/nu14193937

**Published:** 2022-09-22

**Authors:** Justo Sandino, Marina Martín-Taboada, Gema Medina-Gómez, Rocío Vila-Bedmar, Enrique Morales

**Affiliations:** 1Department of Nephrology, Hospital 12 de Octubre, 28041 Madrid, Spain; 2Área de Bioquímica y Biología Molecular, Departamento de Ciencias Básicas de la Salud, Facultad de Ciencias de la Salud, Universidad Rey Juan Carlos, 28041 Madrid, Spain; 3Research Institute of University Hospital 12 de Octubre (imas12), Department of Medicine, Complutense University of Madrid, 28041 Madrid, Spain

**Keywords:** obesity, chronic kidney disease, obesity-related glomerulopathy, visceral adipose tissue, renal sinus adipose tissue

## Abstract

Obesity is recognized as an independent risk factor for the development of kidney disease, which has led to the designation of obesity-related glomerulopathy (ORG). Common renal features observed in this condition include glomerular hypertrophy, glomerulosclerosis, haemodynamic changes and glomerular filtration barrier defects. Additionally, and although less studied, obesity-related kidney disease also involves alterations in renal tubules, including tubule hypertrophy, lipid deposition and tubulointerstitial fibrosis. Although not completely understood, the harmful effects of obesity on the kidney may be mediated by different mechanisms, with alterations in adipose tissue probably playing an important role. An increase in visceral adipose tissue has classically been associated with the development of kidney damage, however, recent studies point to adipose tissue surrounding the kidney, and specifically to the fat within the renal sinus, as potentially involved in the development of ORG. In addition, new strategies for the treatment of patients with obesity-related kidney disease are focusing on the management of obesity. In this regard, some non-invasive options, such as glucagon-like peptide-1 (GLP-1) receptor agonists or sodium–glucose cotransporter-2 (SGLT2) inhibitors, are being considered for application in the clinic, not only for patients with diabetic kidney disease but as a novel pharmacological strategy for patients with ORG. In addition, bariatric surgery stands as one of the most effective options, not only for weight loss but also for the improvement of kidney outcomes in obese patients with chronic kidney disease.

## 1. Introduction

Obesity is now considered a worldwide epidemic, which grows in parallel with the increase in the incidence of chronic kidney disease (CKD) [1]. Thus, there is significant interest in the link between obesity and renal dysfunction. Although it was unclear whether obesity contributed to CKD independently of hypertension and type 2 diabetes (T2D), as these conditions often coexist, it is now accepted that obesity is an independent risk factor for the development of CKD and predicts a higher incidence of progression to ESRD [2]. 

Since an association between obesity and nephrotic-range proteinuria was first described in 1974 [3], several studies have shown that an increase in body mass index (BMI) is directly associated with the risk of developing CKD [4]. In fact, obesity-related glomerulopathy (ORG) is now considered an established disease entity that occurs in obese individuals [5]. Glomerulomegaly and focal segmental glomerulosclerosis (FSGS) are characteristic findings in ORG histopathology. Regarding the etiology, glomerular hyperfiltration, associated with alterations in glomerular hydrostatic pressure, is considered to play a key role in the development of ORG. However, the fact that not all patients with obesity present alterations in renal function suggests that the metabolic complications associated with obesity that trigger renal damage are complex, and that mere weight gain is not a sufficient factor for renal injury [6].

In this regard, how obesity accounts for CKD independently of hypertension and T2D is not clear yet. Typical alterations of adipose tissue occurring during obesity may directly affect renal function [7], including changes in the pattern of adipokine and cytokine secretion by this tissue [8]. In addition, obesity is considered a chronic low-grade inflammatory disease due to the production of proinflammatory cytokines that may trigger the appearance of metabolic alterations [9,10] and may also contribute to the development and/or progression of CKD [11]. Moreover, during obesity, adipose tissue lipid metabolism becomes dysfunctional [12], resulting in increased fatty acid delivery to other peripheral tissues, leading to a situation known as lipotoxicity [13]. Specifically, lipid ectopic deposition in the kidney has been described as one of the mechanisms involved in the development of obesity-related CKD [14]. 

Interestingly, not all adipose tissue depots are the same. In this regard, recent studies have highlighted the potential involvement of the renal sinus fat depot in the impairment of renal function [15]. Identifying the amount, distribution and specific functions of the different fat depots may help elucidate the variability of obese patients and the contribution of specific adipose tissue depots to obesity-related CKD.

## 2. Obesity and Chronic Kidney Disease

### 2.1. Epidemiology and Risk Factors

In recent times, overweight and obesity rates have peaked in terms of proportions worldwide, with estimates as high as 36.9 and 38% in adult men and women, respectively [16]. It has been estimated that overweight and obesity play a key pathogenic role in 15–30% of patients with CKD [17] and act as an independent and strong predictor for new-onset CKD [18]. 

It is important to identify which patients with obesity may develop CKD, since not all of them are at equal risk [19]. It has been widely described that there are different phenotypes of obesity [20]: metabolically healthy obesity phenotype is characterized by high insulin sensitivity, low prevalence of hypertension and a favorable fasting glucose, lipid, and inflammation profile. This group of patients also tends to have more subcutaneous and less visceral fat mass as well as less ectopic fat deposition in the liver, kidney and skeletal muscle than metabolically unhealthy individuals. Conversely, the components of the metabolic syndrome correlate with the typical fat distribution of the metabolically abnormal obese [20], a phenotype associated with a higher risk of incident CKD [6]. On this matter, “renal steatosis” or “fatty kidney” (ectopic lipid accumulation in the kidney) might play a key role in the pathophysiology of ORG, increasing hydrostatic pressure and probably contributing to the impairment of kidney function [7]. 

### 2.2. Clinical Features

To define ORG, proteinuria and an anodyne urinary sediment must be present in a patient with obesity [21,22]. Even in ORG subjects with nephrotic-range proteinuria (>3.5 g/day), full nephrotic syndrome does not develop, and the reason is yet to be elucidated. Among the most accepted hypothesis for this matter is that the gradual progression of proteinuria might allow a compensatory increase in hepatic synthesis of albumin; [23] on the other hand, tubular reabsorption of filtered proteins may be different in the setting of hyperfiltration compared with full nephrotic syndrome-causing glomerulopathies. Murine models have shown that the presence of proteinuria stimulates sodium reabsorption in the distal nephron, which may partially explain the absence of edema in patients with ORG [24]. 

### 2.3. Physiopatology of Obesity-Associated CKD

Growing evidence shows that glomerular hyperfiltration is one of the key elements in ORG etiology. Regardless of the cause, a reduction in functional renal mass derives from adaptive changes in the remaining nephrons, leading to vasodilation of the afferent arteriole, increasing the intraglomerular hydrostatic pressure, which ultimately leads to hyperfiltration [25]. In addition, under physiological conditions, most of the sodium filtered is reabsorbed in the proximal convoluted tubule; in both diabetes and obesity, proximal renal sodium and glucose reabsorption via sodium–glucose cotransporter 1 and sodium–glucose cotransporter 2 (SGLT1 and SGLT2) are increased, this leads as well to the vasodilation of afferent arterioles, with augmented glomerular hydrostatic pressure and glomerular filtration rate (GFR) [25]. Increased hydrostatic pressure in glomerular capillaries causes the glomerular basement membrane (GBM) to enlarge [26,27], resulting in podocyte hypertrophy as an adaptive mechanism. The continuous mechanical challenge on the podocytes finally detaches them from the GBM, leading to podocyturia, enhanced glomerular permeability and the presence of segmental scars typical of FSGS (Figure 1). In other words, a reduced total number of nephrons may lead to glomerular enlargement in patients with ORG. 

Several clinical conditions related with hyperfiltration may be present in one patient at the same time, and it is crucial to know that in ORG subjects exposed to renal mass reduction from any cause the pathogenic mechanisms previously described will be enhanced. An excellent clinical model that allows to illustrate this issue is kidney living donors who develop obesity. The risk of ESKD is higher in overweight or obese patients as compared with the non-obese when exposed to extreme renal mass reduction such as kidney donation [28,29]. Nevertheless, as previously mentioned, many people with obesity do not develop CKD, indicating that obesity per se may not be sufficient to cause kidney damage.

## 3. Role of Adipose Tissue in Obesity-Associated Chronic Kidney Disease

Adipose tissue has long been regarded as a regulator of energy homeostasis, although now it is also considered an important endocrine organ through the secretion of adipokines involved in the regulation of inflammation, insulin response and metabolic homeostasis. Thus, alterations in adipose tissue functionality that occur in the course of obesity, rather than obesity itself, may have an important implication in the development of associated diseases such as ORG (Figure 1) [10,11,30].

### 3.1. Adipose Tissue—Derived Lipotoxicity

The failure in the capacity of adipose tissue to expand to accommodate energy surplus is considered a key mechanism linking obesity to metabolic complications [13]. This impaired adipose tissue expandability, together with an increased sensitivity of the visceral adipose tissue depot to lipolysis, might lead to an overspill of lipids into the circulation that will accumulate ectopically in non-adipose tissues, leading to metabolic abnormalities, a situation known as lipotoxicity [31]. A well-known example of ectopic lipid deposition is non-alcoholic fatty liver disease (NAFLD); however, as aforementioned, ectopic lipid overload also occurs in the kidney, leading to the development of “fatty kidney”, which may constitute a biomarker of obesity-related kidney disease [7,32]. Nevertheless, the contribution of lipotoxicity to obesity-associated kidney disease in humans is difficult to evaluate due to the shortage of non-invasive methods to assess the fatty kidney and the lack of specific protocols and/or established criteria for the radiological diagnosis of fatty kidney in patients with obesity. Even so, renal samples from patients with ORG or metabolically unhealthy obesity have been shown to present extensive lipid accumulation in mesangial cells, proximal tubular epithelial cells and podocytes [7]. Lipid overload causes detrimental effects on the kidney, promoting immune cell infiltration and modulating different signaling cascades, including the activation of pro-inflammatory pathways, endoplasmic reticulum (ER) stress and reactive oxygen species (ROS) production, insulin resistance, lipid metabolism alterations or renin-angiotensin aldosterone system (RAAS) overactivation [14]. 

Specifically, the accumulation of lipids in mesangial cells leads to the acquirement of a foam cell-like phenotype, implying the loss of their migratory capacity and contractile function. Since the proper contraction and relaxation of mesangial cells are critical to preserve glomerular filtration, lipid overload-mediated mesangial cell dysfunction may importantly contribute to renal disease progression [33]. In tubular cells, lipid overload causes mitochondrial dysfunction, with insufficient intracellular fatty acid oxidation as a central pathogenic event mediating renal lipotoxicity in these cells [34]. Lipid overload also induced an increase in inflammatory cytokines and apoptosis [35], all these contributing to extracellular matrix deposition and subsequent tubulointerstitial fibrosis, a final common manifestation of CKD. Moreover, lipotoxicity has been involved in the transition of tubular epithelial cells into myofibroblasts [36], a process known as epithelial-to-mesenchymal transition (EMT), long considered key to the process of tubulointerstitial fibrosis, further confirming a link between disturbances of lipid metabolism and cell function maintenance in tubular cells. An altered lipid metabolism and/or lipid deposition also affects the function and survival of podocytes [37], with disturbed lipid metabolism in these cells playing a critical pathogenic role in ORG [38]. Excessive lipid accumulation in podocytes interferes with the insulin signaling pathway, which is crucial for the proper structure and survival of the podocytes. Hence, lipotoxicity and associated insulin resistance promote inflammation, an increase in ROS production and actin cytoskeleton remodeling, eventually leading to podocyte hypertrophy, detachment and apoptosis [39]. Lipotoxic-induced podocyte loss triggers an adaptive hypertrophic response in remaining podocytes that, when maintained, become maladaptive, leading to overall glomerular growth and development of FSGS [40]. 

Metabolic consequences of ectopic fat accumulation not only depend on the amount of lipid stored but also on the characteristics of the lipid species accumulated [41]. Accordingly, in a recent study from our laboratory, the serum lipidome analysis revealed significant differences in circulating lipid signature in obese patients presenting CKD or not, including an increase in the levels of different triglycerides, diglycerides, phosphatidylethanolamines, phosphatidylcholines and lysophosphatidylcholines in the obese patients with CKD [42]. These data put forward comprehensive mass spectrometry-based lipidomics to measure other lipids aside from conventional lipid profiles as a potential tool to validate possible early markers of risk of CKD in patients with severe obesity. 

### 3.2. Alterations in Adipose Tissue Secretion Pattern

Alterations in the adipose tissue secretion pattern of adipokines and other bioactive molecules that may occur during obesity have also been associated with the pathogenesis of obesity-related metabolic disorders. Different studies have drawn attention to the pathophysiological functions of several adipokines, and their contribution to CKD has been extensively reviewed elsewhere [8,11,30]. For instance, a reduction in adiponectin levels has been related to alterations in glomerular permeability, and thus increased albuminuria due to the fusion of podocyte foot processes [43]. Obesity-associated fat cell hypertrophy has also been related with an increased secretion of angiotensinogen, the precursor of angiotensin II, which may act as a link between obesity and CKD [44,45]. Increased circulating levels of leptin can activate intracellular signaling pathways, such as the TGF-β1-mediated pro-fibrotic pathway, ultimately leading to glomerulosclerosis and proteinuria [30,46]. Along this line, a recent study from our laboratory has shown that the significant decrease in proteinuria, and the improvement in glomerular hyperfiltration observed after bariatric surgery, occurred in parallel with changes in circulating levels of adipose-tissue-derived molecules. In this study, we show a decrease in circulating levels of adipokines and pro-inflammatory cytokines related with obesity-induced metabolic disorders (i.e., leptin), whereas circulating levels of adiponectin, considered protective for kidney function, increased after surgery [47]. Along the same line, another study has shown weight loss and the associated normalization in circulating levels of inflammatory cytokines after bariatric surgery as an effective strategy to reduce hyperfiltration in morbidly obese patients. In addition, this study pointed out the levels of IL-1β before surgery as predictors of hyperfiltration, since hyperfiltration remained unchanged in subjects who did not undergo a reduction in the circulating levels of IL-1β/Caspase-1 [48]. Finally, it is important to remark that obesity is a chronic pro-inflammatory disease. During obesity, adipose tissue is infiltrated with macrophages and other immune cells [49]. Infiltrating macrophages not only expand in number but also polarize from the so-called alternatively activated M2 anti-inflammatory macrophages to the classically activated pro-inflammatory M1 macrophages [50]. M1 macrophages are an important source of angiotensin and multiple pro-inflammatory cytokines [47], which may play a pivotal role in mediating the pathogenesis of obesity-associated metabolic disorders, including the induction and progression of CKD. For instance, an increased secretion of pro-inflammatory cytokines from adipose tissue in obesity may affect the endothelial barrier function by the activation of endothelial cells and immune cells within intrarenal microvessels, eventually leading to irreversible tubular injury, leading to progression to renal fibrosis [11]. Different works have described the pathophysiological functions of many pro-inflammatory cytokines in the context of CKD [8]. In this regard, circulating levels of TNFα and IL-6 have been positively correlated with the incidence of CKD [51]. Moreover, an increase in plasma levels of IL-1 β and caspase 1, which could result from the activation of the inflammasome signaling, has also been described in morbidly obese patients, showing an independent association between eGFR and IL-1β in humans and pointing to a pathogenetic role of the inflammasome signaling in the early stages of nephropathy [48]. 

In addition, pro-inflammatory cytokines also contribute to the development of systemic insulin resistance, which has also been described as an early metabolic alteration in patients with CKD [52]. The effects of pro-inflammatory cytokines together with insulin resistance exacerbate this pathological state, playing an important role in the development of albuminuria and decline of kidney function [53,54].

### 3.3. An Emerging Role for Perirenal Fat and Renal Sinus Fat in Kidney Injury

Despite obesity being already accepted as a risk factor for the development of CKD independently of other metabolic alterations [4], different adipose compartments show distinctive characteristics and differentially influence metabolic risk [55]. In this regard, strong evidence has presented central obesity as a significant risk factor for cardiovascular and metabolic risk, with visceral adipose tissue depot associated with an adverse obesity phenotype, whereas subcutaneous adipose tissue may even be protective [56,57]. However, alterations in organ-specific fat depots occurring during obesity have also been related to cardiometabolic risk. 

In the last years, imaging has facilitated examining smaller fat depots that may interact locally with adjacent tissues. Particularly in the kidney, the deposition of fat may appear in different areas, including the retroperitoneal space, the perirenal area outside the renal capsule, the hilum and the renal sinus [58], with renal sinus fat recently seen as a potential link between obesity and CKD. Specifically, fat accumulation around the renal hilum and sinus has been suggested to increase renal interstitial pressure through the direct compression of vessels exiting the kidney, including arterial and venous blood flow and the nerve bundle, increasing intrarenal venous pressure [15] and pointing to a potential independent association of renal sinus fat accretion with hypertension and CKD [59,60]. However, the exact role in the development of CKD is still controversial. 

Currently, there are some techniques able to delineate perirenal fat, hilum fat, and renal sinus fat, including magnetic resonance imaging and computed tomography. Studies using the latter confirmed that measuring renal sinus fat is possible and reproducible, and positively correlated renal sinus fat with BMI, waist circumference and abdominal visceral fat [61], with several additional cardiometabolic risk factors [62]. More specifically, the ratio of renal sinus fat volume to visceral adipose tissue volume has also been suggested to be an independent risk indicator of coronary artery calcium in patients with coronary artery disease [63,64]. 

Other studies using magnetic resonance imaging associated renal sinus fat with exercise-induced albuminuria in non-diabetic cohorts at diabetic risk (BMI > 27), pointing to this fat depot as a contributor to the pathogenesis of microalbuminuria [65]. Interestingly, renal sinus fat has also been positively correlated with the percentage of intrahepatic fat in individuals with moderate abdominal obesity [66], and the decrease in overall renal sinus fat after a long-term weight loss intervention did not associate with improved renal function or blood pressure parameters but rather with improved hepatic parameters [67]. Accordingly, renal sinus fat has been suggested to have a protective role on renal cells that are disturbed in the presence of NAFLD-derived elevated fetuin-A levels, which mediate renal sinus fat-induced proinflammatory signaling in glomerular cells, thus contributing to the impairment of renal function [68]. Hence, these data point to additional roles for obesity-related renal sinus fat dysfunction, beyond the ectopic deposition of fat in this depot and associated vessel compression.

In addition, in overweight or obese individuals, weight loss has been rather linked to decreases in renal sinus fat, indicating that changes in this fat depot may be metabolically relevant in the kidney [69]. Indeed, renal sinus fat volume is significantly increased in prediabetic subjects and associated with visceral fat mass and cardiovascular risk factors. These data suggest that this tissue may undergo early changes during the development of metabolic disease and may link metabolic disease and associated CKD, even serving as an early biomarker for this disease [70,71]. 

## 4. New Options for the Treatment of Obesity-Associated CKD

### 4.1. Nutritional Management

Nutritional management of obesity consists of a reduced calorie diet, usually complemented with increased exercise, although this approach is yet to show long-term benefits in stopping CKD progression [72]. Most studies show that nutritional management is effective at reducing weight and controlling blood pressure and proteinuria, but long-term effects on eGFR have not been clarified; moreover, no substantial data exist supporting the superiority of a specific dietary pattern to promote weight loss in subjects with CKD [73]. 

Low-carbohydrate, high-protein diets are widely promoted as an effective means for rapid weight loss and better glycemic control, although growing evidence suggests that high-protein diets may be associated with metabolic complications that may be unfavorable to kidney health [74]. As for plant-based diets—the consumption of plant foods with or without small amounts of meat, fish, seafood, eggs, and dairy—the evidence of their effects on the CKD population is scarce [75]. 

### 4.2. New Pharmacological Options

Finerenone is a new non-steroidal mineralocorticoid receptor antagonist. Its role as an agent to treat obesity-related CKD is yet to be stablished, but recent data from the FIDELIO and FIGARO trials found reductions both in cardiovascular events and kidney failure outcomes with finerenone in a selected group of patients [76]. Interestingly, the mean BMI of the patients on finerenone was above 25 kg/m^2^, meaning that the shown benefits of finerenone in the diabetic CKD population are also present in subjects with concomitant obesity. In this line, on murine models, finerenone has shown to have a direct effect activating interscapular brown adipose tissue, representing a promising pharmacologic tool to treat obesity in the setting of CKD, since it is a metabolic disorder associated with adipose tissue dysfunction (Figure 1) [77]. 

Glucagon-like peptide-1 receptor agonists (GLP-1 Ras) have widely demonstrated its beneficial effects on improving both weight control and glycemia [78]. Nevertheless, there has not been any properly designed trial in the CKD population to measure weight loss as a primary outcome [79,80]. In this line, semaglutide (doses of 2.4 mg weekly) is FDA-approved for the management of obesity regardless of the presence of type 2 diabetes, but, again, no solid recommendations can be extracted from landmark trials for patients with concomitant CKD that justify its use in this population, especially on subjects with ESRD [81]. 

On the CKD population, sodium–glucose cotransporter 2 inhibitors (SGLT2i) have shown numerous exceptional properties: decelerating the progression to ESRD in diabetic patients already on RAAS inhibitors (Figure 1) [82], reducing the risk for need of dialysis, transplantation or death due to kidney disease and, extraordinarily, renoprotection across different levels of CKD, even for patients with lower baseline eGFR [82,83]. Despite all these substantial beneficial effects, no SGLT2i is specifically approved for obesity treatment on CKD, despite some reports of modest weight loss with the use of these agents [73]. As for the non-diabetic population, dapagliflozin has shown important protective effects on patients with CKD in terms of cardiovascular and kidney-related mortality. In fact, in the DAPA-CKD trial, non-diabetic subjects on dapagliflozin presented a mean BMI of 29.4 ± 6.0 kg/m^2^ [79], indicating that SGLT2i might play a role in hampering kidney disease progression even in non-diabetic patients with obesity and CKD. 

Other therapies (phentermine–topiramate, bupropion–naltrexone and orlistat) have proven efficacy for achieving weight loss in the general population, but there is no robust data in obese patients with CKD [73]. Recent reports on the use of melatonin suggest that this hormone may exert beneficial effects on lipid profile regulation and insulin resistance, also influencing gut microbiota and inflammation. However, not all preclinical data show evidence of significant weight loss, which makes the role of melatonin in obesity a controversial issue [84].

### 4.3. Bariatric Surgery as a Renoprotective Option

Bariatric surgery (BS) is a well-recognized therapeutic option for severe obesity (Figure 1) with beneficial effects (seen as early as 1 year after surgery) in terms of weight loss and slowing CKD progression. BS has also been associated with reduction in serum creatinine and urinary albumin-to-creatinine ratio, with a significant increase in eGFR in the CKD population from 6 to 24 months after surgery [85]. However, long-term kidney outcomes after BS are rather unclear. Some studies show that, independently of procedure type, after a 10-year median following BS, there was no difference in new-onset albuminuria among groups. Nevertheless, protective effects in terms of reduction in cumulative incidence rates of advanced CKD have been described for surgery recipients compared with control groups, even in the presence of higher base-line albuminuria [86]. The benefits of BS on patients with moderate/severe CKD-related albuminuria can also be seen as soon as 1 year after surgery and, remarkably, the risk of progression to kidney failure can be reduced by 70% at 24 months after BS and by 60% after 5 years [87].

The impact of BS in lipid and metabolic signature in the serum and urine of patients with severe obesity and CKD before and after undergoing BS has been recently described, providing very interesting insights on the identification of new biomarkers and possible future treatment targets [47]. Regarding patients with severe obesity and CKD compared with severely obese patients without CKD, serum lipidome analysis revealed downregulation of levels of triglycerides (TGs) and diglycerides (DGs), as well as a decrease in branched-chain amino acid (BCAA), lysine, threonine, proline and serine. In addition, BS removed most of the correlations in CKD patients against biochemical parameters related to kidney dysfunction [42]. 

## 5. Conclusions

Growing pieces of evidence show that obesity is directly associated with kidney dysfunction. This has led to the definition of the clinical entity known as obesity-related glomerulopathy, which presents with pathophysiological structural and functional features in the kidney. The mechanisms linking both diseases remain elusive, although the hypothesis that alterations in adipose tissue functionality have an impact on renal function is gaining significance. Recent studies point to a connection between lipotoxicity and obesity-associated alterations in the secretion profile of adipose tissue with kidney disease. Moreover, attention is being paid to adipose tissue depots directly related to the kidney, especially renal sinus fat. New CKD therapeutic approaches focus on the treatment of metabolic complications of obesity, such as increased adiposity, poor glycemic control or hemodynamic alterations, among others. Along this line, lifestyle modifications, novel pharmacological treatments and surgical interventions have shown significant benefits and open new avenues for the treatment of obesity-associated kidney disease. 

## Figures and Tables

**Figure 1 nutrients-14-03937-f001:**
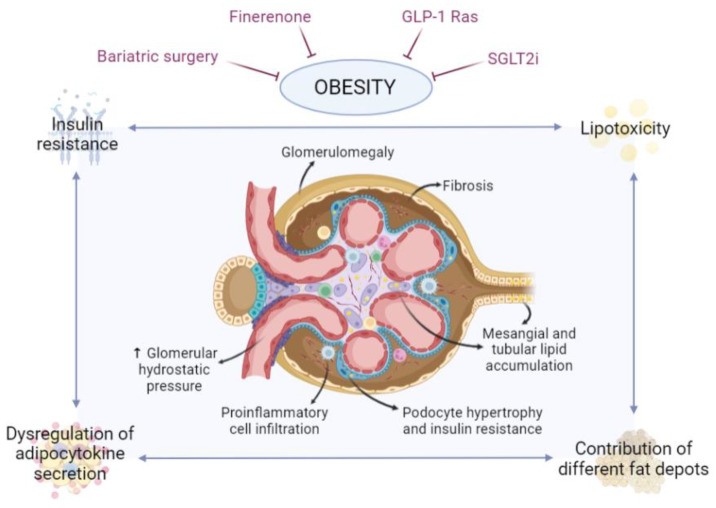
Involvement of adipose tissue in obesity-related glomerulopathy. During obesity, adipose tissue, and specifically certain depots adjacent to the kidney, undergoes changes that may affect the kidney, leading to lipotoxicity, alterations in the pattern of adipokine and cytokine secretion and associated insulin resistance. Effects on the kidney, and specifically on the nephron, may include fat accumulation in mesangial, tubular and podocyte cells, podocyte hypertrophy and insulin resistance, glomerulomegaly, proinflammatory cell infiltration, hemodynamic alterations and fibrosis. Ultimately, this can affect the functionality of the nephron, disrupting the glomerular filtration barrier and potentially leading to the development of chronic kidney disease. There are currently several strategies to combat obesity and, consequently, its deleterious effects, such as kidney damage, including finerenone, GLP-1 Ras, SGLT2i and bariatric surgery, among others. Created with BioRender.com.

## Data Availability

Not applicable.

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
