# Peer review of "Novel Insights in the Physiopathology and Management of Obesity-Related Kidney Disease"

_nutrients, 2022, doi:10.3390/nu14193937_

Round 1
Reviewer 1 Report
This review manuscript well summarized the physiopathology and management of obesity-related kidney disease. In addition, the authors reviewed the importance of fat depot in the kidney. This manuscript gives new information and is well organized for readers.
Minor comment
Please add nutritional management in the section on the treatment of obesity-associated CKD
Page 3, line 101 : correct hiperfiltration to hyperfiltration
Author Response
We thank the reviewer for the opportune recommendations.
- Please add nutritional management in the section on the treatment of obesity-associated CKD: A new section was added (page 6: section 4.1 Nutritional management), with 3 new references (86, 87, and 88)
- Page 3, line 101: correct hiperfiltration to hyperfiltration: Proper correction has been made.
Reviewer 2 Report
In this work, the authors discuss some metabolic complications associated with obesity that trigger renal damage mainly at glomerular and, secondarily, at tubular level. There is an interesting vision about how ectopic fat in kidney may favor the obesity-related glomerulopathy. I think that the manuscript is very well developed and attractive for lectors.
I only have few minor commentaries:
Abstract
- In the abstract is mainly mentioned glomerular alterations due to obesity. However, the authors expose some tubular alteration that are underrepresented in the abstract. I suggest mentioning that in the ORG also tubular alterations may be presented.
- Please, define GLP-1 and SGLT2
3.1 Adipose tissue – derived lipotoxicity
- Line159, please define ER (endoplasmic reticulum) and ROS (reactive-oxygen stress)
- It is recommended to incorporate a recent article in the area, which presents the relation between hyperfiltration and the inflammasome activation in obese patients (DOI: 10.1111/nep.14077). This is important also for co-morbidities such as hypertension.
- Ref 42, please mention the main findings of this article, or the main differences in the lipid signature between CKD and not-CKD obese patients.
3.2 Alterations in adipose tissue secretion pattern
- There is any relation between the cytokine pattern during the obesity and the polarization of macrophages toward a M1 phenotype? Since M1 macrophages have been strongly related with the induction and progression of CKD, it would be recommendable to mention this aspect.
Author Response
Reviewer 2
In this work, the authors discuss some metabolic complications associated with obesity that trigger renal damage mainly at glomerular and, secondarily, at tubular level. There is an interesting vision about how ectopic fat in kidney may favor the obesity-related glomerulopathy. I think that the manuscript is very well developed and attractive for lectors.
I only have few minor commentaries:
Abstract
- In the abstract is mainly mentioned glomerular alterations due to obesity. However, the authors expose some tubular alteration that are underrepresented in the abstract. I suggest mentioning that in the ORG also tubular alterations may be presented:
We appreciate the reviewer for the suggestion and have included a sentence regarding the tubular alterations during obesity (Abstract lines 16-18)
- Please, define GLP-1 and SGLT2
Proper definitions have been included
3.1 Adipose tissue – derived lipotoxicity
- Line 159, please define ER (endoplasmic reticulum) and ROS (reactive-oxygen stress)
Proper definitions have been included
- It is recommended to incorporate a recent article in the area, which presents the relation between hyperfiltration and the inflammasome activation in obese patients (DOI: 10.1111/nep.14077). This is important also for co-morbidities such as hypertension.
We appreciate the reviewer for the bibliographic suggestion due to its importance in the field. We have included two paragraphs in the manuscript regarding this reference. The first one (page 5, lines 214-219) refers to the improvement of inflammatory cytokines after bariatric surgery and highlights the fact that levels of IL-1b prior surgery may be predictor of hyperfiltration. The second one (page 6, lines 234-238) highlights the importance of the activation of inflammasome signaling in the early stages of kidney disease in obese patients.
- Ref 42, please mention the main findings of this article, or the main differences in the lipid signature between CKD and not-CKD obese patients.
We thank the reviewer for raising this point and we have included the main findings of this study in page 5, lines 190-193.
3.2 Alterations in adipose tissue secretion pattern
- There is any relation between the cytokine pattern during the obesity and the polarization of macrophages toward a M1 phenotype? Since M1 macrophages have been strongly related with the induction and progression of CKD, it would be recommendable to mention this aspect.
We agree with this reviewer in the relevance of highlighting this point and have included one sentence drawing attention to the association between obesity, CKD, and the polarization of macrophages towards the pro-inflammatory M1 phenotype (page 5, lines 222-227)